# Synthesis and Properties of Thermally Self-Healing PET Based Linear Polyurethane Containing Diels–Alder Bonds

**DOI:** 10.3390/polym14163334

**Published:** 2022-08-16

**Authors:** Minghui Xu, Ning Liu, Hongchang Mo, Xianming Lu, Jinkang Dou, Bojun Tan

**Affiliations:** 1State Key Laboratory of Fluorine & Nitrogen Chemicals, Xi’an 710065, China; 2Xi’an Modern Chemistry Research Institute, Xi’an 710065, China

**Keywords:** thermally self-healing, PET, linear polyurethane, Diels–Alder reaction

## Abstract

A Diels–Alder (DA) bond containing poly(tetrahydrofuran)-co-(ethyleneoxide) (PET) based linear polyurethane (PET-DA-PU) was synthesized via a prepolymer process using PET as raw material, DA diol as chain extender agent, and toluene-2,4-diisocyanate (TDI) as coupling agent. The structure of PET-DA-PU was characterized by attenuated total reflectance-Fourier transform-infrared spectroscopy (ATR-FTIR), proton nuclear magnetic resonance spectrometry (^1^H NMR) and carbon nuclear magnetic resonance spectrometry (^13^C NMR). The thermal performance and self-healing behavior of PET-DA-PU were investigated by differential scanning calorimetry (DSC), polarized optical microscope, universal testing machine, scanning electron microscopy (SEM) and NMR, respectively. The glass transition temperature of PET-DA-PU was found to be −59 °C. Under the heat treatment at 100 °C, the crack on PET-DA-PU film completely disappeared in 9 min, and the self-healing efficiency that was determined by the recovery of the largest tensile strength after being damaged and healed at 100 °C for 20 min can reach 89.1%. SEM images revealed the micro-cracks along with the blocky aggregated hard segments which were the important reasons for fracture. NMR spectroscopy indicated that the efficiency of retro DA reaction of PET-DA-PU was 70% after 20 min heating treatment at 100 °C. Moreover, the PET-DA-PU/Al/Na_2_SO_4_ composite was also prepared to simulate propellant formulation and investigated by universal testing machine and SEM; its healing efficiency was up to 87.8% under the same heat treatment process and exhibits good self-healing ability. Therefore, PET-DA-PU may serve as a promising thermally self-healing polymeric binder for future propellant formulations.

## 1. Introduction

PET (poly(tetrahydrofuran)-co-(ethyleneoxide)) has attracted considerable attention in recent investigations aimed at developing advanced propellant formulations, owing to its excellent mechanical properties and strong compatibility with energetic materials and nitrate plasticizers [1,2,3]. Therefore, PET was widely used as a polymeric binder in propellant formulations to achieve higher energetic performance and superior mechanical behavior [4,5,6,7]. However, micro-cracks frequently appeared when the propellant suffered from being continuously exposed to the external environment, e.g., mechanical attack, chemical abrasion, and UV radiation [8,9]. These cracks, which are difficult to detect and repair, can adversely affect final properties of the propellant formulations, and even worse, the micro-cracks may develop further into irreversible damage and significantly shorten the service life of propellant formulations [10,11,12]. Smart polymeric binders with self-healing capability are highly desirable for practical application; however, this demand has not been met yet.

In order to produce polymeric materials with more durable security and reliable properties, various strategies were exploited from renewable resources, including encapsulation, hollow fibers, microvascular networks, supramolecular self-assembly, reversible chemistry, dynamic covalent bonds, etc. [13,14,15,16,17,18]. Among them, the Diels–Alder (DA) dynamic covalent bond between furan and maleimide (which needs neither additional healing agent nor catalyst) has garnered increasing attention for synthesizing heat activated healable systems [19,20]. The reason is that the maleimide group shows a relatively high reactivity in the DA reaction, due to the more electron-deficient C=C bond, and the formed DA cycloaddition also could be easily cleaved at 90~150 °C. Hence, the self-healing polymeric materials based on DA reaction have high self-healing efficiency and a repeating ability to heal the damage at the same position.

In the last decades, a variety of self-healing polymers based on DA bonds were reported, and the majority of the research was focused on cross-linked networks, polymer gels and thermosetting polymers [21,22,23]. Typically, the formation of thermally reversible networks based on furan–maleimide DA reaction with maleimide and excess furan were prepared to achieve a satisfying healing efficiency. However, researchers discovered that sufficiently molecular mass polyurethanes were difficult to obtain, because the forward DA reaction gives rise to both endo and exo stereoisomer adducts and does not play a significant role in these macromolecular syntheses, since both participate in the chain growth. Therefore, few teams also concentrated their attention on DA bonds containing linear polyurethane based on furan–maleimide [24,25,26].

In this paper, PET based linear polyurethane with DA bonds (PET-DA-PU) was synthesized using PET as the raw material, DA diol as chain extender agent, and TDI as coupling agent, via a prepolymer process. The chemical structure and thermal performance of polyurethane were characterized by attenuated total reflectance-Fourier transform-infrared spectroscopy (ATR-FTIR), nuclear magnetic resonance spectrometry (NMR) and differential scanning calorimetry (DSC). The self-healing behaviors of as-synthesized PET-DA-PU were demonstrated by polarized optical microscope (POM), universal testing machine, scanning electron microscopy (SEM) and NMR. Moreover, the PET-DA-PU/Al/Na_2_SO_4_ composite was also prepared to simulate the propellant formulation and evaluated by universal testing machine and SEM.

## 2. Experimental

### 2.1. Materials

PET with *M*_n_ of 4000 g mol^−1^ and hydroxyl value of 23.1 mg KOH/g was provided from the Liming Chemical Engineering Research and Design Institute (Luoyang). Dibutyltindilaurate (DBTDL), furfuryl alcohol and 1,1′-(methylenedi-1,4-phenylene) bismaleimide (BMI, 98%) were purchased from Aldrich and used as received. Toluene-2,4-diisocyanate (TDI), N,N-dimethylformamide (DMF), sodium sulfate (Na_2_SO_4_), chloroform, 1,2-dichloroethane and ethanol were supplied by China National Medicines. Aluminium power with the diameter of 20 μm was self-made. All solvents for the reactions were analytical grade and were dried before use.

### 2.2. Synthesis of DA Diol

An amount of 2,2′-(methylenebis(4,1-phenylene))bis(4-(hydroxymethyl)-3a,4,7,7a-tetrahydro-1H-4,7-epoxyisoindole-1,3(2H)-dione) (DA diol) was synthesized via cycloaddition reaction using frufuryl alcohol and 1,1′-(methylenedi-1,4-phenylene) bismaleimide as the raw materials, as shown in Figure 1. Briefly, 1,1′-(methylenedi-1,4-phenylene) bismaleimide (20 g, 55.8 mmol) was dissolved in chloroform (100 mL) in a 250 mL round-bottomed Schlenk flask with a magnetic stirring bar; the reaction solution was heated up to reflux until a uniform system, and then the freshly distilled frufruyl alcohol (11.3 g, 115.2 mmol) was added dropwise into the reaction system. The reaction was allowed to proceed for 30 h. The solution was then poured into 8 fold excess of isopropanol to precipitate the crude product. The precipitation process recrystallization in isopropanol/DMF mixed solvent three times, and the DA diol was obtained after drying under vacuum at 60 °C as a pale yellow solid (22.3 g, yield: 72%).

### 2.3. Synthesis of PET-DA-PU

PET based polyurethane with self-healing properties based on Diels–Alder reaction, PET-DA-PU, was synthesized via the successive prepolymerization of PET using TDI as the curing agent, and DA diol as the chain extender, which is illustrated in Figure 2. In a typical synthesis example, exactly anhydrous PET (20 g, 5.0 mmol) and freshly distilled 1,2-dichloroethane (30 mL) were placed in a 250 mL three-neck round-bottomed Schlenk flask with a mechanical stirrer, condenser and thermometer under N_2_ atmosphere, and then heated to 60 °C; TDI (3.48 g, 20 mmol) and DBTDL (20 µL) dissolved in 20 mL 1,2-dichloroethane were added drop by drop into the reaction solution. After stirring for 2 h, DA diol (8.32 g, 15 mmol) was added into the reaction mixture, which was then stirred at 80 °C for an additional 3 h. After the polymerization reaction was finished, the reaction system was precipitated by adding to excess ethanol under vigorous stirring. The product PET-DA-PU was finally obtained after drying under vacuum at 40 °C. (30.97 g, yield: 97.4%).

### 2.4. Synthesis of PET-DA-PU/Al/Na_2_SO_4_ Composite

The PET-DA-PU/Al/Na_2_SO_4_ composite was synthesized with a weight ratio of 1/0.5/0.5. Briefly, 20 g of PET-DA-PU was dissolved in ethyl acetate (30 mL) in a 200 mL beaker, Al power (10 g) and Na_2_SO_4_ (10 g) were added into the solution and the mixture was then left under intensive stirring for 1 h. The mixture was then poured into the Teflon-mold, and the solvent in mixture was removed under vacuum. Finally, the PET-DA-PU/Al/Na_2_SO_4_ composite was obtained after being dried in vacuum at 40 °C for 24 h.

### 2.5. Self-Healing Property

The self-healing property of PET-DA-PU was evaluated by three means: (1) The healing process of PET-DA-PU was observed under POM equipped with temperature programmed heating stage. Briefly, ‘X’ crack with 0.5 mm in depth of PET-DA-PU films was cut with a thin knife, while a sample with a crack was put on the heating stage at 100 °C. The photographs were taken at different time intervals during the healing process. (2) Evaluating the self-healing efficiency PET-DA-PU films quantitatively by tensile test. In a typical example, the well-prepared PET-DA-PU film was cut off, and heated at 100 °C for 20 min, then cooled down to 60 °C and kept for 48 h to obtain the healed PET-DA-PU film. The tensile properties of pristine and healed films were tested on a universal testing machine, and the self-healing efficiency was calculated by the recovery of the largest tensile strength. (3) NMR spectroscopy was employed to study the thermal reversibility of PET-DA-PU and investigate the degradation rate of DA bonds in PET-DA-PU under heating treatment.

### 2.6. Characterization

Chemical structure: the chemical structure of PET-DA-PU was examined by ATR-FTIR and NMR. ATR-FTIR was performed on a Bruker Tensor 27 instrument (KBr pellet) in the range 4000–400 cm^−1^. NMR spectra were performed on a Bruker 500 MHz instrument with CDCl_3_ as the solvent at 25 °C and tetramethylsilane as the internal standard. Thermal property: the thermal property of PET-DA-PU was investigated by differential scanning calorimetry (DSC). DSC was performed in a TA Instruments Q1000 equipment in the range of −100~0 °C at a heating rate of 10 °C min^−1^ under a nitrogen flow. POM (Zeiss Axioskop 2 plus) with a heating stage was employed to observe the evolution of cracks on PET-DA-PU film. Mechanical property: AG-X Plus universal testing machine (Shimadzu, Kyoto, Japan) was used to measure the tensile property of PET-DA-PU films in accordance with GB/T528-1998 with a loading rate of 500 mm min^−1^. The specimens were cut into dumbbell-like with the sizes of 20 mm (neck length) × 4 mm (width) × 2 mm (thickness) and kept at 0% humidity for 7 days before measurement. Samples were conducted independently in quintuplicate, and the results were presented with the average values. Fracture morphology of the films which were prepared by freeze-fractured (films were frozen in the liquid nitrogen and ruptured) and tensile-fractured were observed on a Tescan Vega 3 LMU scanning electron microscope (Tescan, Brno, Czech Republic).

## 3. Result and Discussion

### 3.1. Preparation and Characterization of PET-DA-PU

PET-DA-PU was synthesized via a prepolymer process of PET, using DA diol which prepared though cycloaddition reaction as chain extender, and TDI as a curing agent, as demonstrated in Figure 1 and Figure 2. The chemical structure of as-synthesized PET-DA-PU was first investigated by ATR-FTIR (as shown in Figure 1). In the ATR-FTIR of PET-DA-PU, the shoulder peaks at 1775 and 1513 cm^−1^ specific were due to the characteristic absorption of DA ring on PET-DA-PU [27]. The peak at 1107 cm^−1^ is ascribed to the characteristic absorption of C–O from PET. As compared to PET, the intensity decreases in –OH peak (3490 cm^−1^) of PET and the absence of adsorption peak at around 2270 cm^−1^ indicated that there were no residual hydroxyl and isocyanate groups in PET-DA-PU. It is worth noting the additional peaks at 3295, 1714 and 1540 cm^−1^ which were due to the –NH stretching vibration, –C=O band and –N–H stretching bands of the urethane group on PET-DA-PU, respectively [28]. Therefore, it could be confirmed that PET based polyurethane containing DA bonds had been synthesized successfully.

The chemical structure of as-synthesized PET-DA-PU was also characterized by ^1^H NMR and ^13^C NMR (as shown in Figure 2). The peaks at 1.61, 3.46 and 3.62 ppm were assigned to the methylene protons of PET as shown in Figure 2a, and the corresponding carbon atom signals appearing at 26.6 and 70.3 ppm in Figure 2b. Meanwhile, the peaks at 3.04, 3.15, 5.15 and 6.54 ppm due to DA ring on the PET-DA-PU. The signals at 1.25 and 7–8 ppm were attributed to the methyl protons and hydrogen protons on the benzene rings of TDI, respectively, and the corresponding carbon signals appeared at 16.9 ppm and 127–135 ppm [29]. All of the results above confirmed that PET-DA-PU has been synthesized successfully via the prepolymerization of PET.

### 3.2. Glass Transition Temperature of PET-DA-PU

Glass transition temperature (*T*_g_) is an important property of polymeric binders as it determines the application temperature range. In this study, DSC was used to investigate the *T*_g_ of PET-DA-PU, and the *T*_g_ was measured by the cooling DSC curves of the PET-DA-PU. As shown in Figure 3, in comparison of PET prepolymer (*T*_g_ = −84 °C), the PET-DA-PU showed a slightly higher *T*_g_ of −59 °C, which allows it to work in a low-temperature environment. It is due to the decrease in chain flexibility of PET as to the introduction of DA diol and TDI, and the formation of urethane group [30]. Moreover, PET-DA-PU exhibits only one *T*_g_, which may be attribute to the fact that PET and TDI-extended DA diol segments are well-dispersed and blended within the films, thus the polyurethane consists of one phase. The same phenomenon was also observed in other literature [31].

### 3.3. Self-Healing Property of PET-DA-PU

The self-healing property of PET-DA-PU under heat treatment was firstly investigated by observing the healing process with POM, as shown in Figure 4 [32]. The PET-DA-PU film sample with 1 mm thickness was prepared, and an ‘X’ crack with 0.5 mm in depth was produced by a thin knife on the film. The PET-DA-PU film sample was put on the heating stage and heated for 100 °C and observed by POM.

As shown in Figure 4, the width of the crack in PET-DA-PU film decreased clearly with the heat treatment time increasing, and the crack on the film had completely disappeared in 9 min [33]. The results indicated good self-healing properties of PET-DA-PU.

The healing efficiency of the PET-DA-PU was measured quantitatively by the universal testing machine [34]. The PET-DA-PU film samples were prepared and tested according to GB/T528-1998. The PET-DA-PU film was cut off by a thin knife, and the film was heated at 100 °C for 20 min, then cooled down to 60 °C and kept for 48 h, and the healed PET-DA-PU film was obtained. As shown in Figure 5, compared with the PET-DA-PU (the tensile strength was 0.92 MPa, with an elongation at 151.3%), the tensile strength of repaired PET-DA-PU was 0.82 MPa with an elongation at 137.5%. Thus, the healing efficiency of PET-DA-PU was 89.1%. The results reveal that the PET-DA-PU has a satisfactory healing efficiency.

To investigate the rupture mechanism of PET-DA-PU in the tension process, the fracture morphologies of the PET-DA-PU films which were prepared by freeze-fractured and tensile-fractured were studied by SEM. As shown in Figure 6a,b, the wrinkles are marked by red restangular frames representative of aggregated hard segments, and the ravines are marked by blue circle frames representative of non-homogeneous soft segments. Clearly, the phase separation exists between the soft and hard segments and is well dispersed within the whole vision [35,36,37,38]. However, the hard segments (marked as red rectangular frames) in tension-fractured PET-DA-PU film display aggregated as blocky shapes in Figure 6c,d, and the micro-cracks along with the blocky aggregated hard segments which were the important reasons for fracture were also observed [39]. These results revealed the fracture of hard segments and the formation of micro-cracks resulting in the fatigue fracture of PET-DA-PU film.

### 3.4. Thermal Reversibility of PET-DA-PU

To study the thermal reversibility of PET-DA-PU, NMR was employed to investigate the changes in ^1^H NMR spectroscopy after the heat treatment in Figure 7 [40]. It is well known that BMI and furan prepolymer formed in the retro-DA reaction of PET-DA-PU under heat treatment. As marked in Figure 7, the chemical shift at 3.17 ppm (labeled as a), 4.93 and 4.96 ppm (labeled as b), 6.54 ppm (labeled as c), 6.63ppm (labeled as d), 5.24 ppm (labeled as e), 7.14 ppm (labeled as g), 7.35 ppm (labeled as f) decreased visibly, and the new chemical shifts at 7.25 ppm (labeled as a’), 7.28 ppm (labeled as f’), 7.15 ppm (labeled as g’) owing to BMI, and chemical shifts at 7.69 ppm (labeled as b’), 6.46 ppm (labeled as c’), 6.54 ppm (labeled as d’), 5.09 ppm (labeled as e’) owing to furan prepolymer appeared after heat treatment [41,42,43,44]. It is indicated that DA bonds in PET-DA-PU de-bonded under heat treatment.

To investigate the efficiency of retro DA reaction under heat treatment quantitatively, the changes in ^1^H NMR spectroscopy of DA diol were also studied [45]. As shown in Figure 8, the chemical shifts at 5.18 ppm (labeled as a), 4.08 and 3.77 ppm (labeled as b), 6.57 ppm (labeled as c), 5.0 ppm (labeled as d), 3.02 and 3.18 ppm (labeled as e), 7.35 ppm (labeled as f), 7.14 ppm (labeled as g) owing to DA diol were significantly decreased after heat treatment, and the new chemical shifts at 5.19 ppm (labeled as a’), 4.38 ppm (labeled as b’), 6.27 and 6.38 ppm (labeled as c’), 7.57 ppm (labeled as d’) owing to frufuryl alcohol, and 7.36 ppm (labeled as e’), 7.26 ppm (labeled as f’), 7.15 ppm (labeled as g’), 4.03 ppm (labeled as h’) owing to BMI were observed. Based on the changes in the ^1^H NMR spectroscopy peak areas of DA diol after heat treatment, conversion x can be calculated from the following formula:(1)$$x=\frac{A_{c\prime }+A_{c\prime \prime }}{A_{c}+A_{c\prime }+A_{c\prime \prime }}$$
A_c_, A_c’_, A_c’’_ integral areas of peaks c, c’, c’’, respectively.

According to the formula, the conversion of DA diol was 70% after 20 min heating treatment at 100 °C. It demonstrated that DA bonds in PET-DA-PU can degrade into BMI and furan prepolymer in a short time via retro DA reaction, and it is good benefit for repairing of PET-DA-PU. All of these results indicate that PET-DA-PU has a good thermal reversibility.

### 3.5. Healing Behavior of PET-DA-PU Based Composites

To evaluate the healing efficiency of PET-DA-PU in propellant, the PET-DA-PU/Al/Na_2_SO_4_ composite was prepared and measured by the universal testing machine. The PET-DA-PU/Al/Na_2_SO_4_ composite was prepared in a ratio of 1/0.5/0.5, and the obtained composite film was cut into dumbbell-shaped specimens and tested according to GB/T528-1998. The PET-DA-PU/Al/Na_2_SO_4_ composite film was cut off and heated at 100 °C for 20 min, then cooled down to 60 °C and kept for 48 h and the healed PET-DA-PU/Al/Na_2_SO_4_ composite films were obtained. As shown in Figure 9 and Table 1, in comparison with the PET-DA-PU/Al/Na_2_SO_4_ composite (the tensile strength was 0.82 MPa, with an elongation at 138.8%), the tensile strength of repaired PET-DA-PU/Al/Na_2_SO_4_ composite was 0.72 MPa with an elongation at 113%. Hence, the healing efficiency of PET-DA-PU/Al/Na_2_SO_4_ composite was 87.8%, which was slightly lower than PET-DA-PU. It is due to the reduction in DA bond density in composite as to the introduction of Al and Na_2_SO_4_ and decrease the healing efficiency of the PET-DA-PU/Al/Na_2_SO_4_ composite. However, the PET-DA-PU/Al/Na_2_SO_4_ composite still has a high healing efficiency exceeding 85% and exhibits outstanding self-healing performance.

The fracture morphologies of the PET-DA-PU/Al/Na_2_SO_4_ composite which were prepared by freeze-fracturing and tensile-fracturing were also observed by SEM. As shown in Figure 10a,b, the crack surface of the composite was smooth and the interface between the exposed particles and polymeric matrix was fuzzy, meaning that the interface adhesion property between solid filler and matrix is strong. However, a number of exposed particles and cracks appeared in Figure 10c,d, and voids were found between the exposed particles and polymeric matrix, meaning that the interface adhesion property between solid fillers and PET-DA-PU matrix became poor during the tensile process [46]. Generally, the cracks and voids also occurred in aging stages of polymeric binder. Therefore, the PET-DA-PU polymeric binder which can repair the micro-cracks within the matrix under heat treatment could prolong its service life. 

## 4. Conclusions

A Diels–Alder bond containing PET based linear polyurethane, PET-DA-PU, was synthesized using PET as raw material, DA diol as chain extender agent, and TDI as coupling agent. From ATR-FTIR, ^1^H NMR and ^13^C NMR results, the PET-DA-PU was synthesized successfully via prepolymer process. The DSC curves indicated that PET-DA-PU had a low glass transition temperature of −59 °C, which allows it to work in a low temperature environment. The evolution of cracks on PET-DA-PU was observed by POM, and the results indicated that the cracks on PET-DA-PU film completely disappeared in 9 min under the heat treatment at 100 °C. A tensile test was used to determine the self-healing efficiency by the recovery of the largest tensile strength after being damaged and healed; the self-healing efficiency of PET-DA-PU can reach 89.1% after 20 min heating treatment at 100 °C. NMR spectroscopy indicated that the efficiency of retro DA reaction of PET-DA-PU can reach up to 70% after 20 min heating treatment at 100 °C. SEM images were used to investigate the fracture morphologies of the PET-DA-PU film, and the results revealed the micro-cracks along with the blocky aggregated hard segments which were the important reasons for fracture. Moreover, the PET-DA-PU/Al/Na_2_SO_4_ composite was also prepared to simulate the propellant formulation, and its healing efficiency was 87.8% under the same heat treatment. In the SEM images of tensile-fractured PET-DA-PU/Al/Na_2_SO_4_ composite, exposed particles, cracks and voids were observed, meaning poor interface adhesion property between solid fillers and PET-DA-PU matrix. Consequently, PET may find its application as a novel self-healing binder in propellant formulations.

## Data Availability

The data presented in this study are available on request from the corresponding author.

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
