# Peer review of "Synthesis and Properties of Thermally Self-Healing PET Based Linear Polyurethane Containing Diels–Alder Bonds"

_polymers, 2022, doi:10.3390/polym14163334_

Round 1

Reviewer 1 Report

Change throughout the text: "Diel-Alder" into "Diels-Alder"

Enter the correct name (as per IUPAC recommendation) of the PET

Abstract:

provide the full names: ATR-FTIR, 1HNMR, 13CNMR

Experimental

Toluene diisocyanate - what isomer (2,4- or 2,6-)? Provide an appropriate name and change the TDI marking

The work is written correctly, the introduction, experimental and discuusion is appropriate. Summary and references also without comments. In general, the text is valuable, I gave my little remarks at the beginning.

Author Response

Q1. Change throughout the text: "Diel-Alder" into "Diels-Alder"

A: According to the expert’s suggestion, we have revised the mistake in the revised manuscript.

Q2. Enter the correct name (as per IUPAC recommendation) of the PET

A: According to the expert’s request, the name of PET was corrected in the revised manuscript.

Q3. Abstract: provide the full names: ATR-FTIR, 1HNMR, 13CNMR

A: According to the expert’s suggestion, we have provided the full names in the revised manuscript.

Q4. Experimental: Toluene diisocyanate - what isomer (2,4- or 2,6-)? Provide an appropriate name and change the TDI marking

A: According to the expert’s request, we have provided the appropriate name of TDI which was also demonstrated in Scheme 2 in the revised manuscript.

Q5. The work is written correctly, the introduction, experimental and discuusion is appropriate. Summary and references also without comments. In general, the text is valuable, I gave my little remarks at the beginning.

A: We appreciate the reviewer’s patient review and careful comment of our manuscript. According to the expert’s suggestion, we have revised the errors in terminology, and the whole manuscript was proofread by an expert in the field.

Reviewer 2 Report

Comments to authors are listed below:

1.      Abstract must be enriched via valuable results which pave the way for understanding the audiences.

2.      The introduction section is short and poorly described. It does not present the reference to the manuscript scope. In the introduction section, Authors should make an in-depth literature review concerning the application of PET applications. So, Introduction has deficiency citation to valuable works published before such as:

https://link.springer.com/article/10.1007/s12034-018-1587-1

https://www.mdpi.com/2073-4360/13/9/1471

https://journals.sagepub.com/doi/10.1177/0892705711412816

3.      DSC curves in Figure 3 should discuss in detail and compare with previous literature to present the thermal improvements.

4.       Conclusion is very short and lack the basic fundamentals of the results obtained. Please, authors should re-write the conclusions again  with more emphasis on the significant comparison and the improvements from the results obtained.    

Author Response

Q1. Abstract must be enriched via valuable results which pave the way for understanding the audiences.

A: We appreciate the reviewer’s patient review and careful comment of our manuscript. According to the expert’s suggestion, we have improved the Abstract in the revised manuscript.

Q2. The introduction section is short and poorly described. It does not present the reference to the manuscript scope. In the introduction section, Authors should make an in-depth literature review concerning the application of PET applications. So, Introduction has deficiency citation to valuable works published before such as: https://link.springer.com/article/10.1007/s12034-018-1587-1, https://www.mdpi.com/2073-4360/13/9/1471, https://journals.sagepub.com/doi/10.1177/0892705711412816

A: We have improved the introduction section in revised manuscript. We have carefully read these given works, and find that the Polyethylene terephthalate (PET) in these works is different from our study (poly(tetrahydrofuran)-co-(ethyleneoxide), PET), but another valuable literatures were cited as Reference 4-6 in revised manuscript.

Q3. DSC curves in Figure 3 should discuss in detail and compare with previous literature to present the thermal improvements.

A: According to the expert’s request, we have rewritten the discussion of Figure 3 in the revised manuscript, and Reference 30 was added for comparison.

Q4. Conclusion is very short and lack the basic fundamentals of the results obtained. Please, authors should re-write the conclusions again with more emphasis on the significant comparison and the improvements from the results obtained.   

A: According to the expert’s request, we have rewritten the Conclusion in the revised manuscript.

Reviewer 3 Report

In this contribution, the authors prepare a PET-DA-PU copolymer containing Diels-Alder (DA) bonds. The PET-DA-PU copolymer exhibits a self-healing capability. However, improvements in design of experiment, data analysis and discussion are necessary to support the conclusion. Following questions and comments need to be addressed before making further decisions.

1. The hydroxyl value is defined as the number of milligrams of potassium hydroxide required to neutralize the acetic acid taken up on acetylation of one gram of the sample (Pure Appl. Chem., 1973, Vol. 33, No. 2-3, pp 411-436). Line 67 reports a hydroxyl value of 0.7%, more likely to be the weight fraction of hydroxyl groups in PET. Given the Mn of 4000 g/mol for PET, the actual hydroxyl value should be approximately 28.

2. What are the degrees of crystallinity of PET-DA-PU before and after reparation? Different thermal histories may result in different crystallinity, furthermore, different tensile strength before and after reparation. In addition, the DSC curve of PET-DA-PU in Figure 3 shows an exothermic signal near 0 °C, what transition is this signal related to?

3. In Figure 6, the wrinkles are assigned as aggregated hard segments, and ravines as soft segments. But, are the micrometer-scale aggregates actually formed by the phase separation of this random copolymer? This needs further proof, such as FTIR mapping, EDS, heavy-metal staining, etc. 

4. In Figure 7, why Peak e appears as a doublet? Besides, Peak c is at a lower field than Peak d, whereas Peak c’ is at a higher field than Peak d’ after the retro Diels-Alder reaction. Can the authors justify why this shift takes place?

5. In line 270, the authors claim that a lower DA bond density of PET-DA-PU/Al/Na2SO4 results in a lower healing efficiency than neat PET-DA-PU. How do the authors justify the contribution of DA bond to self-healing? First, the scission of polymer chains, if any, may not take place at DA bonds, so nor does the healing. Instead, the breaking and healing of dumbbell bars are more attributed to sliding/de-entanglement and re-entanglement of polymer chains. Second, is the self-healing easier/faster for a high fraction of PET-DA-PU than that of a low fraction of PET-DA-PU as in the composite? Maybe not, either. Third, even a polymer without DA bonds can also gain segment mobility to heal cracks at a temperature way above Tg. Does the retro DA reaction in PET-DA-PU promote a faster self-healing due to the higher mobility of dissociated segments? Can this be approved by, for example, rheology?

Author Response

Q1. In this contribution, the authors prepare a PET-DA-PU copolymer containing Diels-Alder (DA) bonds. The PET-DA-PU copolymer exhibits a self-healing capability. However, improvements in design of experiment, data analysis and discussion are necessary to support the conclusion. Following questions and comments need to be addressed before making further decisions.

The hydroxyl value is defined as the number of milligrams of potassium hydroxide required to neutralize the acetic acid taken up on acetylation of one gram of the sample (Pure Appl. Chem., 1973, Vol. 33, No. 2-3, pp 411-436). Line 67 reports a hydroxyl value of 0.7%, more likely to be the weight fraction of hydroxyl groups in PET. Given the Mn of 4000 g/mol for PET, the actual hydroxyl value should be approximately 28.

A: We appreciate the reviewer’s patient review and careful comment of our manuscript. According to the expert’s suggestion, the hydroxide value with 23.1 mg KOH/g was given in the revised manuscript. It is notice that the actual hydroxide value (23.1) was lower than the theory hydroxide value (28). It may due to the end-capping by catalyzer (e.g. boron trifluoride etherate) in the cationic polymerization of tetrahydrofuran and ethyleneoxide, and also to the production of outgrowth of crown ether.

Q2. What are the degrees of crystallinity of PET-DA-PU before and after reparation? Different thermal histories may result in different crystallinity, furthermore, different tensile strength before and after reparation. In addition, the DSC curve of PET-DA-PU in Figure 3 shows an exothermic signal near 0 °C, what transition is this signal related to?

A: This is a good question. We make the same heat treatment to ensure the equally state of PET-DA-PU before and after reparation. However, the degrees of crysallinity of PET-DA-PU are not the main factor affecting its self-healing property, and were not investigate furthermore in this study. In addition, the original DSC curves of PET-DA-PU and PET prepolymer were shown in Figure 1-2. The DSC cooling curves were opted to estimate the glass transition temperature of polymer. Obviously, the exothermic signal near 0 °C was due to the unstable baseline during the initial period of cooling process, and the phenomena appeared both in Figure 1 and Figure 2 (see attachment PDF ).

Q3. In Figure 6, the wrinkles are assigned as aggregated hard segments, and ravines as soft segments. But, are the micrometer-scale aggregates actually formed by the phase separation of this random copolymer? This needs further proof, such as FTIR mapping, EDS, heavy-metal staining, etc.

A: This is a good question. In this study, the SEM images of freeze-fractured PET-DA-PU gels were investigated. The micrometer-scale aggregates formed by the phase separation of polymer were explained by other literatures (Polymers 2020, 12, 603-616, and Materials 2019, 12, 1932-1953), which were also cited as Reference 32-33 in the revised manuscript. Unfortunately, we couldn’t give the further proof soon, but we will resolve the issue in the future.

Q4. In Figure 7, why Peak e appears as a doublet? Besides, Peak c is at a lower field than Peak d, whereas Peak c’ is at a higher field than Peak d’ after the retro Diels-Alder reaction. Can the authors justify why this shift takes place?

A:  It’s a mistake, we have corrected it in Figure 7 in the revised manuscript.

Q5. In line 270, the authors claim that a lower DA bond density of PET-DA-PU/Al/Na2SO4 results in a lower healing efficiency than neat PET-DA-PU. How do the authors justify the contribution of DA bond to self-healing? First, the scission of polymer chains, if any, may not take place at DA bonds, so nor does the healing. Instead, the breaking and healing of dumbbell bars are more attributed to sliding/de-entanglement and re-entanglement of polymer chains. Second, is the self-healing easier/faster for a high fraction of PET-DA-PU than that of a low fraction of PET-DA-PU as in the composite? Maybe not, either. Third, even a polymer without DA bonds can also gain segment mobility to heal cracks at a temperature way above Tg. Does the retro DA reaction in PET-DA-PU promote a faster self-healing due to the higher mobility of dissociated segments? Can this be approved by, for example, rheology?

A: This is a very good question, but it is actually extremely difficult to answer the question definitively. In this study, we investigated the self-healing efficiency in PET-DA-PU/Al/Na2SO4 to simulate the propellant formulation, and we got positive results. However, as far as we know, the discussion about the detailed influence between DA reaction and mobility of dissociated segments has not been reported. However, the good idea would stimulate us to search some method to resolve the issue in the future.

Round 2

Reviewer 2 Report

Comments to authors are listed below:

Authors should include another paragraph explaining what the PET polymer and its properties and applications. So, Introduction has deficiency citation to valuable works published before such as:

https://link.springer.com/article/10.1007/s12034-018-1587-1,

https://www.mdpi.com/2073-4360/13/9/1471,

https://journals.sagepub.com/doi/10.1177/0892705711412816

All peaks of NMR in Figure 7 should be clarified and discuss in detail.

Regarding, mechanical properties are illustrated in Figure 9 Authors should include a table, summarising the storage modulus and strain at break of all PET composites, instead of presenting stress-strain curve.

Author Response

Q1. Authors should include another paragraph explaining what the PET polymer and its properties and applications. So, Introduction has deficiency citation to valuable works published before such as: https://link.springer.com/article/ 10.1007/s12034-018-1587-1, https://www.mdpi.com/2073-4360/13/9/1471, https://journals.sagepub.com/doi/10.1177/0892705711412816.

A1: We appreciate the reviewers taking the time to review our manuscript again. According to the expert’s requirement, the valuable works were cited as Reference 4-6 in revised manuscript.

Q2. All peaks of NMR in Figure 7 should be clarified and discuss in detail.

A2: According to the expert’s suggestion, the peaks of NMR in Figure 7 were clarified and discussed in revised manuscript.

Q3. Regarding, mechanical properties are illustrated in Figure 9 Authors should include a table, summarising the storage modulus and strain at break of all PET composites, instead of presenting stress-strain curve.

A3: According to the expert’s requirement, table 1 including elastic modulus and elongation at break of all PET composites was illustrated in revised manuscript.

Reviewer 3 Report

I appreciate the authors’ point-to-point response and revisions of the manuscript. Besides my expectations of the resolutions of Question 3 and 5 in future investigations, I have two more comments:

1. Following Question 4 regarding Figure 7, both Proton d and d’ tend to show a doublet shape due to the 3J coupling. Whereas, Proton c and c’ show a more complex shape (e.g., J. Lab.Chem. Educ. 2019, 7, 8-18doi:10.5923/j.jlce.20190701.02Quím. Nova 202144, 1-14. doi.org/10.21577/0100-4042.20170639). Can the authors confirm the assignments in Figure 7 are compliant with this experience?

2. Line 311, ATR-FTIR instead of ART-FTIR.

Line 55, “abundantly of self-healing polymer” seems not appropriate.

Author Response

Q1. I appreciate the authors’ point-to-point response and revisions of the manuscript. Besides my expectations of the resolutions of Question 3 and 5 in future investigations, I have two more comments:

  1. Following Question 4 regarding Figure 7, both Proton d and d’ tend to show a doublet shape due to the 3J coupling. Whereas, Proton c and c’ show a more complex shape (e.g., J. Lab.Chem. Educ. 2019, 7, 8-18. doi:10.5923/j.jlce.20190701.02; Quím. Nova 2021, 44, 1-14. doi.org/10.21577/0100-4042.20170639). Can the authors confirm the assignments in Figure 7 are compliant with this experience?

A1: We appreciate the reviewers taking the time to review our manuscript again. According to the expert’s requirement, we have clarified the peaks of NMR in Figure 7 carefully. As to the complex composition in polymeric materials, the NMR peaks of large molecules polymer always show a complex shape, which is obviously different from the small molecules (e.g., Figure 8), the same phenomenon also illustrated in other literatures (e.g., Journal of Applied Polymer Science 2015, 132, 41944-41952, Polymer 2018, 142, 33-42). Besides, the given valuable works were cited as Reference 43-44 in revised manuscript.

Q2. Line 311, ATR-FTIR instead of ART-FTIR.

A2: The mistake was corrected in revised manuscript.

Q3. Line 55, “abundantly of self-healing polymer” seems not appropriate.

A3: According to the expert’s suggestion, we have rewritten the inappropriate sentence in the revised manuscript.